# Complexes Between Adamantane Analogues B_4_X_6_ -X = {CH_2_, NH, O ; SiH_2_, PH, S} - and Dihydrogen, B_4_X_6_:*n*H_2_ (*n* = 1–4)

**DOI:** 10.3390/molecules25051042

**Published:** 2020-02-26

**Authors:** Josep M. Oliva-Enrich, Ibon Alkorta, José Elguero

**Affiliations:** 1Instituto de Química-Física “Rocasolano”, CSIC, Serrano, 119, E-28006 Madrid, Spain; 2Instituto de Química Médica, CSIC, Juan de la Cierva, 3, E-28006 Madrid, Spain; ibon@iqm.csic.es (I.A.); iqmbe17@iqm.csic.es (J.E.)

**Keywords:** hydrogen storage, boron, noncovalent interactions, quantum chemistry

## Abstract

In this work, we study the interactions between adamantane-like structures B_4_X_6_ with X = {CH_2_, NH, O ; SiH_2_, PH, S} and dihydrogen molecules above the Boron atom, with ab initio methods based on perturbation theory (MP2/aug-cc-pVDZ). Molecular electrostatic potentials (MESP) for optimized B_4_X_6_ systems, optimized geometries, and binding energies are reported for all B_4_X_6_:*n*H_2_ (*n* = 1–4) complexes. All B_4_X_6_:*n*H_2_ (*n* = 1–4) complexes show attractive patterns, with B_4_O_6_:*n*H_2_ systems showing remarkable behavior with larger binding energies and smaller B···H_2_ distances as compared to the other structures with different X.

## 1. Introduction

Hydrogen storage is becoming an important issue regarding the energetic needs of our modern world [1,2]. Different methods are designed for hydrogen storage [3,4,5], including cryogenics, high pressures, and chemical compounds that reversibly release dihydrogen upon heating [6]. Among the different systems described in the literature, the metal organic frameworks (MOF) have been the most successful ones as hydrogen storage [7,8,9,10,11].

Related to the computational study in the chemical trapping of dihydrogen, noncovalent interactions must be taken into account [12], given the relatively weak attracting force—mostly dispersive—derived from two neutral molecules, leading to general complexes with formula Z:*n*H_2_, where Z is a neutral molecule and *n* is the number of H_2_ molecules attached to the Z neutral system. A number of theoretical articles have been devoted to the interaction of dihydrogen with metallic systems [13,14,15,16,17,18,19,20,21,22].

The adamantane scaffold is widely used due to their steric [23,24], lipophilic [25] and rigid characteristics [26,27,28,29,30]. Derivatives that include heteroatoms have been synthesized, in addition to the well-known aza-(AZADO and hexamethylenetetramine) [31,32,33,34] and oxo-derivatives (tetrodotoxin) [35]. Other derivatives involving C/As/O (arsenicin A) [36], C/N/S (tetramethylenedisulfotetramine) [37], P/S (phosphorus pentasulfide) [38], P/N [39], and C/P/S [40] have also been described.

For the study of dihydrogen complexes, we propose the use of adamantane analog systems where each CH tetrahedral vertex in adamantane is substituted by a Boron atom, and the remaining CH_2_ moieties is substituted by divalent X groups, with X = {CH_2_, NH, O; SiH_2_, PH, S} thus leading to B_4_X_6_ tetrahedral molecules, as shown in Scheme 1. 

The B_4_X_6_:*n*H_2_ complexes (*n* = 1–4) could be formed by approaching H_2_ molecules towards the B atom centered vertically along the four equivalent local Ĉ_3_ rotation axis. The existence of adamantane analogs B_4_X_6_ is not known, except for the 1-boraadamantane, where only one tetrahedral CH vertex is substituted by a B atom [41], with the remaining structure unaltered; this system has also been the target for a computational study for complex formation [42] with Lewis acids and superacids. Further substitutions of B atoms in tetrahedral sites have been studied from a theoretical point of view only [43]. Directly related to this work is the concept of the, σ-hole, proposed by Politzer and Murray [44,45,46], which refers to the electron-deficient outer lobe of a *p* orbital involved in a covalent bond, especially when one of the atoms is highly electronegative, that present positive values for the electrostatic potential [47,48]. On the other hand, when the deficient outer lobe of a *p* orbital involved in a covalent bond is perpendicular or axially oriented with respect of the molecular frame, the electrostatic nature of the interactions considered between B atoms in B_4_X_6_ systems and H_2_ molecules can be rationalized in terms of π-holes [49]. We should emphasize the relation between the Lewis acidity of trivalent B centers and the (non)planarity of the structure surrounding the B atom [50].

## 2. Results and Discussion

### 2.1. Molecular Electrostatic Potential (MESP) and π-Holes in B_4_X_6_ Systems

As stated above, we can rationalize the electrostatic nature of the interaction between B_4_X_6_ and H_2_ molecules in terms of π-holes, namely regions of positive electrostatic potential perpendicular or axially-oriented (as in the adamantane structure) with respect to a portion of the molecular framework, as shown in Figure 1. The empty/electron-deficient *p* lobe of Boron pointing outwards in the B_4_X_6_ systems is an electron (or surplus charge density) attractor, as shown by the positive values of the π-holes.

As shown in Figure 1b, the MESP of the B_4_O_6_ system shows areas of positive (blue) and negative (red) values corresponding to deficient electron density (negative charge attractor) and surplus electron density (positive charge attractor) areas, respectively. Clearly, the electron density deficiency area above the Boron atoms in B_4_X_6_ could attract the electron density of the σ bond of the H_2_ molecule. As shown in Figure 1b, the large electronegativity of the three oxygen atoms bound to boron must have a stronger effect on the attachment of H_2_ molecules as a function of the π-hole values:

π-hole (au): 0.131 (O) >> 0.058 (NH) > 0.047 (SiH_2_) > 0.034 (CH_2_) > 0.025 (PH, S).

### 2.2. Geometries and Energies of B_4_X_6_:nH_2_ Complexes (n = 1–4)

Figure 2 shows the optimized geometries of the isolated B_4_X_6_ systems at MP2/aug-cc-pVDZ level, corresponding to energy minima for all cases. The cartesian coordinates for the B_4_X_6_ optimized structures are gathered in Appendix A, with the MP2 method and basis sets aug-cc-pVDZ and aug-cc-pVTZ, of double-ζ and triple-ζ quality respectively, including diffuse and polarization functions.

In the computations, we first obtain the energy profile of a frozen H_2_ molecule approaching this B atom (*d* distance) along the corresponding local C_3_ axis, as shown in Figure 3 for the B_4_X_6_:H_2_ complexes.

From Figure 3 we can clearly observe that all energy profiles are attractive for an H_2_ molecule down to 3 Å, and then three different curve patterns emerge: (i) for X = {CH_2_, NH, PH, S} the energy profile becomes repulsive when *d* < 3 Å (ii) for X = O, the energy minimum well is flatter and becomes repulsive shifting down to values of *d* ~ 1.7–2.0 Å; and finally (iii) for X = SiH_2_ the energy profile remains attractive down to 1.25 Å. The inset plot of Figure 3—upper right corner—shows an energy profile zoom-in of the region 2.5 Å < *d* < 3.3 Å in order to see more clearly the positions of the energy minima regions, for a given X. Clearly, the {CH_2_, NH}, and {PH, S} curves show similar energy minima regions: We turn from an attractive to a repulsive system at *d* ≤ 2.1 Å (CH_2_), 2.2 Å (NH), 2.48 Å (PH), and 2.55 Å (S). As stated above, a zoom-in of the energy profile for 2.5 Å ≤ *d* ≤ 3.3 Å is included in order to unveil the effect of approaching a H_2_ molecule to the B_4_X_6_ system where several curves have similar profiles. If we observe closely the curves from the zoom-in inset of Figure 3, the energy minima for CH_2_, NH, PH, and S are located as follows: *d*_min_(CH_2_) ~ 2.73 Å, *d*_min_(NH) ~ 2.77 Å, *d*_min_(PH) ~ 3.05 Å, *d*_min_(S) ~ 3.06 Å. For O and SiH_2_ there are no minima within this region since the curves are always attractive.

Once we choose the *d* which corresponds to the energy minimum in Figure 3, we relax the nuclear coordinates in the whole complexes hence determining the energy minimum structure for the B_4_X_6_:*n*H_2_ systems. Due to the different behavior of the B_4_(SiH_2_)_6_ system versus an H_2_ molecule—permanent attractive profile for *d* down to 1.25 Å—as compared to the other complexes—Figure 3—and the lack of an energy minimum geometry for the B_4_(SiH_2_)_6_:H_2_ complex – a geometry optimization shows a bond breaking in the H_2_ molecule and a rearrangement of the B_4_(SiH_2_)_6_ adamantane structure—this system will be analyzed further in another work. The optimized structures for all B_4_X_6_:*n*H_2_ complexes (*n* = 1–4) are depicted in Appendix A, except for B_4_O_6_:*n*H_2_ (*n* = 1–4), the latter shown in Figure 4. In Table 1 we gather the average B···H_2_ and H···H distances in the optimized geometries of the different B_4_X_6_:*n*H_2_ complexes, all corresponding to energy minima at the MP2/aug-cc-pVDZ level of theory.

As regards to the B···H_2_ distances (*d*) in the B_4_X_6_:nH_2_ complexes, as gathered in Table 1, three groups can be clearly distinguished: (i) X = {CH_2_, NH} with *d* distances of ~ 2.7 Å (ii) X = {PH, S} with *d* distances of ~ 3.0 Å and (iii) X = O, with shorter *d* distances down to ~ 1.6 Å. The case for B_4_O_6_ is quite remarkable. As more H_2_ molecules are attached to B_4_O_6_, the *d*(B···H_2_) distances are elongated steadily up to *d* ~ 1.85 Å, and the H···H molecules remain slightly stretched down to Δ ~ 0.016 Å. This behavior for the B_4_O_6_ systems is unique as compared to the other systems since in the latter the attachment of the H_2_ molecules is quite farther to the B atom and the H_2_ molecules remain practically unaltered. There is no clear tendency—as compared to B_4_O_6_—for the *d* distances as more H_2_ molecules are added for X = {CH_2_, NH, PH, S}, with tiny differences for the series 1 ≤ *n* ≤ 4.

Turning now to the H-H distances in the complexes, as shown in Table 1, in the energy minimum structures of the complexes, the H-H distances are very similar as compared to the isolated H_2_ molecule, 0.755 Å. However, there is an exception for the oxygen complexes: when one H_2_ molecule is attached to the B_4_O_6_ system, the H···H bond is elongated by Δ ~ 0.02 Å. As further H_2_ molecules are attached to the B_4_O_6_ system, this elongated H-H bond is shortened consecutively by ~ 0.004 Å, down to 0.763 Å in each H-H molecule of the B_4_O_6_:4H_2_ complex, though still 0.008 Å longer than in the isolated H-H molecule.

Finally, we show the computed binding energies of the H_2_ molecules for the different complexes B_4_X_6_:*n*H_2_ (*n* = 1–4), as seen in Table 2 and displayed in Figure 5, where we also include the CBS extrapolated values. As expected from the computed MESP and energy profiles in B_4_X_6_:H_2_ complexes, the larger binding energy for one H_2_ molecule corresponds to the B_4_O_6_ system, with ΔE ~ 29 kJ/mol (ΔE_CBS_ ~ 22 kJ/mol). For comparative purposes, the electronic binding energy of the water dimer is ΔE[(H_2_O)_2_] ~ 21 kJ/mol [51].

The maximum binding energy for the complexes corresponds to B_4_O_6_:4H_2_ with a value of 79 kJ/mol (CBS extrapolation 60 kJ/mol). However, the binding energy of one H_2_ molecule attached to the other B_4_X_6_ systems is remarkably smaller in comparison, especially when the CBS extrapolation is added. The addition of more H_2_ molecules to the complexes shows practically additive relations for all X. As displayed in Figure 5, when extrapolated to the CBS limit, we can see several features regarding the binding energies in B_4_X_6_:*n*H_2_ (*n* = 1–4) complexes as compared to the MP2/aug-cc-pVDZ energies: (1) the CBS extrapolated binding energies are smaller for a given X and *n* (2) the (absolute value of the) slope of ΔE versus *n* (number of H_2_) molecules decreases for CBS extrapolated values (3) both CBS extrapolated and non-extrapolated binding energies follow a similar linear trend, except for X = O, the latter with clearly larger (CBS) binding energies, from 20 kJ/mol (*n* = 1) to 60 kJ/mol (*n* = 4).

We should notice that for X = O, though the CBS extrapolated slope is smaller than the non-extrapolated one, yet this slope is larger (in absolute value) as compared to the other Xs hence the peculiar behavior of B_4_O_6_:*n*H_2_ as compared to complexes with different Xs. We should also emphasize the small differences (less than ~ 5 kJ/mol ) between binding energies for different Xs for a given number *n* of attached H_2_ molecules, with the exception of X = O with larger binding energies.

## 3. Computational Method

All geometries of the B_4_X_6_ systems and the corresponding B_4_X_6_:*n*H_2_ (*n* = 1–4) complexes were optimized with second-order Møller-Plesset perturbation theory (MP2) [52] and a double-ζ basis set including polarization and diffuse functions [53], such as aug-cc-pVDZ. The interactions between B_4_X_6_ systems and H_2_ molecules are clearly of noncovalent nature, weaker than conventional chemical bonds, given the closed-shell nature of the species involved and the lack of any further singlet coupling between unpaired electrons. We search for stable complexes B_4_X_6_:*n*H_2_ and dispersive corrections are important given the neutral and spin-zero nature of the involved systems, hence the use of MP2 theory in computations. This theory improves on the Hartree–Fock (a mean-field—molecular-orbital—theory of electronic structure) method by adding electron correlation effects by means of Rayleigh–Schrödinger perturbation theory (RS–PT). 

The quantum-chemical computations in this work were carried out at the MP2 level of theory with the scientific software Gaussian09 (Gaussian Inc, Wallingford, CT, USA) [54], and the molecular electrostatic potential (MESP) for the B_4_X_6_ systems was computed with the DAMQT program [55,56], also at the MP2 level of theory. Frequency computations were performed in order to check the energy minimum nature in all B_4_X_6_ systems and B_4_X_6_:*n*H_2_ complexes (*n* = 1–4). The binding energies for the B_4_X_6_:*n*H_2_ complexes are computed as ΔE = E(B_4_X_6_:*n*H_2_) – E(B_4_X_6_) – *n*·E(H_2_), and reported in kJ/mol. Further geometry optimizations of all B_4_X_6_ complexes were carried out with the MP2/aug-cc-pVTZ computational model—with a triple-ζ basis set—in order to check the validity of the optimized geometries (see Appendix A). A single-point energy profile (MP2) versus B···H_2_ distance in the complex B_4_(CH_2_)_6_:H_2_ was computed using both basis sets: aug-cc-pVDZ and aug-cc-pVTZ (see Appendix A), double-ζ and triple-ζ respectively. As shown in Appendix A, the results show similar profiles along the local Ĉ_3_ axis of rotation on the B atom and therefore we can confirm the validity of the MP2/aug-cc-pVDZ computational model for geometries and binding energies. 

In order to assess the dependency of the binding energies on basis set incompleteness, we also computed the binding energies of all complexes in the extrapolated complete basis set (CBS) limit. The CBS energy has been calculated by extrapolation of the HF energies calculated at aug-cc-pV*k*Z, with *k* = D, T and Q, and Equation (1) and the correlation part with Equation (2). The sum of the two components (HF and correlation) (Equation (3)) provides the MP2(CBS) energy.
(1)$$E_{HF,k}=E_{HF,lim}+Ae^{-Bk}$$
with *k* = 2, 3 and 4 for aug-cc-pVDZ, aug-cc-VTZ and aug-cc-pVQZ basis sets, respectively [57,58].
(2)$$E_{coor,k}=E_{corr,lim}+Ak^{-3}$$
with *k* = 3 and 4 for aug-cc-VTZ and aug-cc-pVQZ basis sets, respectively [59]. Finally, we have
(3)$$E_{MP2}\left(CBS\right)=E_{HF,lim}+E_{corr,lim}$$

## 4. Conclusions

From the results obtained in this work we can conclude with the following points:

1) The MESP in the adamantane-like structures B_4_X_6_, with X ={CH_2_, NH, O ; SiH_2_, PH, S}, show π-holes above the B atom with electron (density) attraction forces largest for B_4_O_6_ and lowest for B_4_(PH)_6_ and B_4_S_6_.

2) The energy profiles of one H_2_ molecule approaching along a *C_3v_* axes the B atom of B_4_X_6_ systems show attractive patterns up to certain values for all systems where it turns to repulsive below ~ 2.1 Å for B_4_(CH_2_)_6_ and B_4_(NH)_6_ and below ~ 2.5 Å for B_4_(PH)_6_ and B_4_S_6_, except for B_4_(SiH_2_)_6_ where the profile is always attractive with H_2_ bond breaking and cage rearrangement. For B_4_O_6_, there is a flat energy minimum region within 1.7–2.0 Å.

3) The attraction strength for electron density towards boron atoms in B_4_X_6_ is also shown in the energy profiles of the B_4_X_6_:H_2_ complexes as a function of the B···H_2_ distance *d*. The *d* distances in the energy minima structures coincide with the predicted distances from the energy profiles.

4) The binding energies of the B_4_X_6_:*n*H_2_ complexes—*n* = 1–4, follow a similar linear additive pattern (in magnitude and direction) for all X, except X = O, with larger binding energies. CBS extrapolation shows a significant decrease in the binding energies.

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
