# Peer review of "Complexes Between Adamantane Analogues B4X6 -X = {CH2, NH, O ; SiH2, PH, S} - and Dihydrogen, B4X6:nH2 (n = 1–4)"

_molecules, 2020, doi:10.3390/molecules25051042_

Round 1
Reviewer 1 Report
In this submission, the authors propose adamantane-like structures B4X6 with X = {CH2, NH, O ; SiH2, PH, S} and computationally modeled their interactions with dihydrogen molecules in the context of hydrogen storage. According to the authors, the MP2/aug-cc-pVDZ methods predicted larger binding energies and smaller B···H2 distances for the B4O6:nH2 system as compared to the other systems.
1. Change the notation to cite the references: p1, L17 “storage,3-5“ instead of “storage,3,4,5“
Author Response
Answer: Unfortunately it is not possible with the current format to change the notation of the references. We hope that this can be done when editing the manuscript at the editorial office.
Reviewer 2 Report
This paper describes complexes adamantine analogues B4X6 with dihydrogen molecule. Binding energies between them were estimated by theoretical calculations using MP2/cc-pVDZ and cc-pVTZ level of theory. The origin of this type of interaction is originated with non-covalent bonds and binding energies are so small, about 10 kJ/mol. In such systems, basis set superposition error (BSSE) should become comparable in the estimated energy. At least, BSSE must be estimate for B4X6 with H2 complexes. This will be a good estimation for larger complexes such as B4X6:nH2 (n>2).
From the above points, the author should add the above data in the revised manuscript. I cannot recommend this paper is accepted to the journal of Molecules at the present form.
Author Response
Answer: We have computed the BSSE for the complexes B4X6:H2 by means of the counterpoise correction, at the level of theory MP2/aug-cc-pVDZ and obtained the following values:
BSSE(B4(CH2)6:H2) = 0.001731437370 au = 4.6 kJ/mol
BSSE(B4(NH)6:H2) = 0.001413983811 au = 3.7 kJ/mol
BSSE(B4O6:H2) = 0.004141954436 au = 10.9 kJ/mol
BSSE(B4(PH)6:H2) = 0.001761032935 au = 4.6 kJ/mol
BSSE(B4S6:H2) = 0.001948920224 au = 5.1 kJ/mol
When adding the BSSE to the interaction energy we obtain a corrected complexation energy close to zero. However, it is well known that the counterpoise method overestimates the correction and the problem is even more serious at correlated levels like MP2. For this reason we have added the following sentence at the end of Section 2 “Computational method”, lines 93-102 on page 3:
“The use of ab initio supermolecule calculations is known to be susceptible of Basis Set Superposition Error (BSSE) when finite basis sets are used. The most common way to correct the BSSE is with the full counterpoise method.[i] Systematic studies at the RHF level have indicated that the counterpoise corrected interaction energies are no more reliable than the uncorrected ones.[ii] At correlated levels, the application of the full counterpoise method can cause a nonphysical increase in the dimension of the virtual space[iii] that produces an overestimation of the correction.[iv],[v] Since, the inclusion of diffuse function has been shown to markedly reduce the BSSE effect,[vi],[vii] the interaction energy of the clusters in the present article has been calculated as the difference between the supermolecule and the sum of the isolated monomers in their energy minimum configuration.” The added references here [i-vii] correspond to references [56-62] in the revised manuscript.
[i] S. F. Boys, F. Bernardi, Mol. Phys. 1970, 19, 553-566.
[ii] D. W. Schwenke, D. G. Truhlar, J. Chem. Phys. 1985, 82, 2418-2426.
[iii] D. B. Cook, J. A. Sordo, T. L. Sordo, Int. J. Quantum Chem. 1993, 48, 375-384.
[iv] J. A. Frey, S. Leutwyler, J. Phys. Chem. A 2005, 9, 6990-6990.
[v] J. A. Frey, S. Leutwyler, J. Phys. Chem. A 2006, 10, 12512-12518.
[vi] M. J. Frisch, J. E. Del Bene, J. S. Binkley, H. F. Schaefer III, J. Chem. Phys. 1986, 84, 2279-2289.
[vii] B. F. King, F. Weinhold, J. Chem. Phys. 1995, 103, 333-347.
Reviewer 3 Report
The authors analyzed the interaction between boron cluster B4X6 and dihydrogen theoretically, which are informative for theoretical and experimental chemists having interest in dihydrogen storage chemistry.
Therefore, I recommend publication in Molecules. However, the following point should be considered before publication.
The energy profile for H2 addition to B4(SiH2)6 showed a very different tendency from other substituent systems. What happens in this system ? Does B-Si bond cleavage occur ?
I ask the author to add some comment about this point, otherwise the reader may feel some frustration.
Author Response
Answer: When optimizing the geometry of the complex (B4(SiH2)6):H2 the result is very puzzling, with the H-H broken and the adamantane structure arranged in a completely different geometry – in fact the optimized goemetry is not a (B4(SiH2)6):H2 complex strictly speaking. As mentioned in the original manuscript this deserves a further study. See lines 204-206
Round 2
Reviewer 2 Report
Along with our reviewer’s comments, the authors re-considered binding energies between B4X6 with hydrogen molecule with BSSE correction using the counterpoise method. But obtained binding energies become negative and showed these two molecules did not bind. They added several comments that BSSE by the counterpoise method may be possible to overestimate that effect and other discussion. Then they decided their new results did not include in the manuscript. But we have to forget the whole present target systems are still unknown compounds and the experimental binding energies have not been measured. The author’s data show that H2 is included in B4X6 without the counterpoise correction, while H2 is not in B4H6 with. The above results show that the binding energies for the present target system are too small to estimate them using the author’s method correctly. It is common that BSSE is significant in weakly interacting system. At the present computational level of theory, they cannot discuss chemical properties of these compounds. More high-level sophisticated calculations should be needed.
From the above points, I decide this paper is rejected to the journal of Molecules.
Author Response
We have included the BSSE corrected energies using the Complete-Basis-Set (CBS) extrapolation method as shown in Section 2 (lines 80-89), modified Table 2 and created new Figure 4 where CBS extrapolated and non-extrapolated energies are shown for comparative purposes. We have modified the discussion text accordingly lines 224-264. We agree with the referee that BSSE are important in weak interacting systems and we have reflected this issue with the aboce modifications and with the final sentence in the conclusions. In this particular case, the CBS extrapolated energies show smaller binding energies and less "additive" effect (the slope in Figure 4, in absolute valie, is smaller in CBS extrapolated values).